# The Diagnosis and Evolution of Patients with LARS Syndrome: A Five-Year Retrospective Study from a Single Surgery Unit

**DOI:** 10.3390/cancers16244175

**Published:** 2024-12-14

**Authors:** Cosmin Vasile Obleagă, Sergiu Marian Cazacu, Tiberiu Ștefăniță Țenea Cojan, Cecil Sorin Mirea, Dan Nicolae Florescu, Cristian Constantin, Mircea-Sebastian Șerbănescu, Mirela Marinela Florescu, Liliana Streba, Dragoș Marian Popescu, Ionică Daniel Vîlcea, Mihai Călin Ciorbagiu

**Affiliations:** 1Department of Surgery, University of Medicine and Pharmacy of Craiova, 200349 Craiova, Romania; cosmin.obleaga@umfcv.ro (C.V.O.); ionica.vilcea@umfcv.ro (I.D.V.); mihai.ciorbagiu@umfcv.ro (M.C.C.); 2Department of Gastroenterology and Hepatology, University of Medicine and Pharmacy of Craiova, 200349 Craiova, Romania; sergiu.cazacu@umfcv.ro (S.M.C.); nicolae.florescu@umfcv.ro (D.N.F.); cristian.constantin@umfcv.ro (C.C.); 3Department of Medical Informatics, University of Medicine and Pharmacy of Craiova, 200349 Craiova, Romania; mircea.serbanescu@umfcv.ro; 4Department of Pathology, University of Medicine and Pharmacy of Craiova, 200349 Craiova, Romania; mirela.florescu@umfcv.ro; 5Department of Oncology, University of Medicine and Pharmacy of Craiova, 200349 Craiova, Romania; liliana.streba@umfcv.ro; 6Department of Extreme Conditions Medicine, University of Medicine and Pharmacy of Craiova, 200349 Craiova, Romania; dragos.popescu@umfcv.ro

**Keywords:** low anterior rectal resection, rectal cancer, LARS

## Abstract

Low anterior resection syndrome (LARS) encompasses a multitude of gastrointestinal symptoms that arise secondary to low and very-low anterior rectal resection performed for rectal cancer. This syndrome can affect a significant percentage of patients and impacts life for varying periods or may become permanent, requiring conversion to a permanent colostomy. For this reason, LARS has begun to be studied and recognized by surgeons and oncologists, and patients must be informed about the possibility of developing this condition after curative surgery. The study was conducted on 102 patients diagnosed and operated on for upper, middle, and lower rectal neoplasm, where resection was accompanied by partial or total mesorectal excision, and who exhibited at least one symptom associated with LARS. Obesity, the size of the remaining rectum, total mesorectal excision, anastomotic complications, and prolonged ileostomy time are convergent factors in the etiology of LARS.

## 1. Introduction

Low anterior rectal resection syndrome encompasses multiple gastrointestinal symptoms occurring after low and very-low resection of the rectum for rectal cancer, mainly due to postoperative anatomical changes of the rectum and the posterior perineum [1]. In patients with cancer of the middle and lower rectum, resection associated with total excision of the mesorectum aims to remove the portion of the rectum comprising the tumor and restore digestive transit with the preservation of the sphincter apparatus, thus avoiding a permanent colostomy [2]. Up to 80% of the patients can develop secondary digestive symptoms to these types of resections [1,2,3], which are defined as LARS syndrome that impairs quality of life over a variable period or is sometimes permanent, requiring conversion to definitive colostomy [2,3]. For this reason, anterior resection syndrome (LARS) began to be studied and acknowledged by surgeons and oncologists; thus, patients must be informed about the possibility of its appearance after curative surgical intervention, and treatment should be instituted as early as possible [4,5,6].

Low anterior resection syndrome (LARS) includes the development of various digestive symptoms following anal sphincter-preserving rectal surgery. According to the consensus definition, the patient must exhibit at least one of the following symptoms: variable or unpredictable bowel function, altered stool consistency, increased stool frequency, pain during defecation, evacuation difficulties, urgency, incontinence, or involuntary leakage. Additionally, these symptoms must result in at least one of the following consequences: toilet dependency, constant concern about bowel function, unsatisfactory bowel function, strategies or compromises related to defecation, impact on mental and emotional well-being, effect on daily social activities, impact on social relationships, or interference with social roles and responsibilities [1,2,3,4,5,6,7]. Factors such as obesity, diabetes, type of surgery, and the length of the preserved rectum may influence the appearance of LARS symptoms [1,2].

The article presents a retrospective, unicentric, and descriptive study on the diagnosis of LARS in patients who underwent surgeries for rectal cancer, its causes, and evaluation of treatments, thus providing significant data for a full understanding of this syndrome moving forward.

## 2. Materials and Methods

We conducted a retrospective study that included all patients admitted to the General Surgery Section II of the County Clinical Emergency Hospital in Craiova between 1 October 2017 and 1 September 2022 and diagnosed with neoplasm of the upper, middle, and lower rectum, in which a low-anterior-type resection of the rectum was performed, with partial or total excision of the mesorectum, depending on the location of the ileostomy. We assessed the demographic data, clinical history, clinical and imaging (CT; MRI) aspects of the type of neoadjuvant treatment, surgical technique, duration of ileostomy, results of anatomopathological analysis, evaluation of anastomosis by colonoscopy at 6 weeks, onset and duration of LARS-specific symptoms, and reassessment of LARS symptoms 2 years after surgery.

The inclusion criteria were as follows: patients diagnosed with upper, middle, and lower rectal cancer in which a type of previous resection was performed; with R0 resection assessed by histopathology; and evaluated at 6 weeks by colonoscopy. In the case of ileostomy, the closure was performed in patients without indication for postoperative adjuvant treatment. Patients received questionnaires with LARS-specific symptoms upon resumption of bowel transit (or after the dissolution of the ileostomy) and then again at 6 months and 2 years. Patients with distant metastasis, those who died immediately postoperatively, those with *p* with R1 or R2 resection or local recurrence, and those with missing data were excluded. The diagnosis of rectal carcinoma was confirmed in all cases by colonoscopy with anatomopathological examination and also by MRI evaluation (which aided in the staging and topography of the tumor). All patients had colonoscopic and CT or MRI reevaluation after neoadjuvant therapy. In cancers of the upper rectum, lower anterior resection (LAR) with partial excision of the mesorectum (the final anastomosis was below the level of peritoneal reflection) was performed, while in tumors of the middle and lower rectum, a low and ultra-low anterior resection (ULAR) with total excision of mesorectum (TME) (colorectal anastomosis was up to 2 cm from the anorectal junction) was accomplished. A LARS questionnaire was administered in all cases, with points allocated for the flatus lack of control (0–7 points), incontinence for liquid stools (0–3 points), frequency of bowel movements (0–5 points), clustering of stools (0–11 points), and urgency (0–16 points). The diagnosis of LARS was established according to the current consensus by 2020 diagnostic criteria [7] and included patients with a score between 0 and 42 (Table 1), encompassing, respectively, non-LARS (score 0–20), minor LARS (LARS score 21–29), and major LARS (LARS score ≥ 30).

The first line of treatment consisted of a diet with high fiber associated with antidiarrheal drugs, and in the case of persistence of symptoms, high-volume enemas (above 250 mL) were recommended together with physical therapy exercises.

The Ethics Committee of the Craiova County Emergency Clinical Hospital was informed, and this study (35514/13.08.2024) was approved on the following bases: (1) The data were collected in a retrospective, observational, descriptive, and non-experimental study; (2) the study did not interfere with current medical care; (3) no substances were administered to the patients, and no biological samples were collected in the study; and (4) the data were collected and analyzed anonymously so that the confidentiality of patient data was not violated.

The following variables were collected: (1) basic characteristics (age, gender, and comorbidities); (2) characteristics of rectal cancer (location, tumor stage, neoadjuvant therapy, type of surgery, type of anastomosis, and complications at local postoperative periods: anastomotic stenosis and anastomotic disjunction (pelvic fistula/abscess)); (3) colonoscopic reassessment of the anastomosis at 6 weeks; and (4) LARS assessment (LARS-specific symptoms, their duration, and maintenance or remission of LARS symptoms at 6 months and 2 years after surgery). The stage of the tumor was assessed by using the 8th edition of the “TNM Classification of Malignant Tumors” from the International Union for the Control of Cancer [8]. Patients were divided into three groups according to the presence or absence of LARS (minor or major).

For statistical analysis, Microsoft Excel 2019 MSO (version 2304 Build 16.0.16327.20200) was utilized to construct a comprehensive database incorporating all variables of interest. MedCalc statistical software (version 20.218) was employed for detailed statistical evaluations. Frequencies were expressed as absolute counts and percentages. Comparisons of ordinal or nominal variables were conducted using chi-square tests. Continuous variables were analyzed using analysis of variance (ANOVA), provided they were normally distributed. A *p*-value of <0.05 was considered indicative of statistical significance. The strength of association between two quantitative variables was assessed using Pearson’s correlation coefficient (r). Correlation strength was classified as moderate for r-values between +0.4 and +0.69 or −0.4 and −0.69 and as strong for r-values exceeding +0.7 or falling below −0.7 [9].

## 3. Results

We included 120 patients with carcinoma of the upper, middle, and lower rectum, in which a low-anterior-type resection of the rectum was performed, with partial or total excision of the mesorectum, depending on the location of the ileostomy (Table 2). A flow diagram of the patient-selection process is presented in Figure 1.

The group of patients with LARS syndrome included 24 patients (23.5%) with minor LARS and 36 patients with major LARS (35.3%). The control group included 42 patients (41.2%).

Regarding comorbidities, the presence of diabetes mellitus and obesity were significantly correlated with the presence of major and minor LARS. A total of 54 patients from the entire group were obese (IMG greater than 30 kg/m^2^), from which 28 patients had major LARS, 8 had minor LARS, and 12 patients had no LARS (*p*-value < 0.001). Of the 38 patients with diabetes mellitus (type I and II) with LAR, 22 developed major LARS and 10 minor LARS (*p*-value < 0.001). We found a higher frequency of LARS in men (83.3% in the major LARS group and 53.3% in the minor LARS group) compared to women (*p*-value 0.03) (Figure 2).

The location of the tumor represents the main criteria for the type of resection; in upper rectal tumors, partial resection of the mesorectum is recommended, while for middle and lower rectal cancers, total excision of the mesorectum is advised. In our study, we included 28 patients with upper rectal cancer, 44 with middle rectal cancer, and 30 with lower rectal neoplasm (Figure 3). The LARS rate was significantly higher in patients with low resection and total mesorectal excision compared with other interventions; therefore, in low and very-low resections for medium and lower rectal cancers, major LARS is much more common (18 vs. 16 vs. 2) (*p*-value < 0.00001).

In patients requiring neoadjuvant therapy, we found no significant correlation between chemotherapy/radiotherapy and the development of postoperative LARS (*p*-value 0.086), but in patients with a locally advanced stage and a significant reduction in size after radiotherapy (Figure 4, Figure 5 and Figure 6), the LARS rate was higher. We found major LARS in 32 patients who underwent radiotherapy versus 2 patients without radiotherapy, minor LARS in 22 patients who underwent radiotherapy versus 4 without, and 26 patients with radiotherapy classified as non-LARS versus 10 patients without radiotherapy and non-LARS.

We noted statistically significant differences regarding LARS presence between the type of anastomosis (more frequent in mechanical vs. manual; *p*-value 0.007); however, we must take into account that middle and lower rectal types benefit more frequently from mechanical anastomosis. The occurrence of postoperative complications such as anastomotic disjunction (pelvic abscess/anastomotic fistula) and stenosis is significantly correlated with the development of LARS (*p*-value 0.04). Of the 16 patients with postoperative complications (15.6%), 12 presented LARS, of which 10 showed major LARS (7 needed a colostomy after more than 2 years from the tumor surgery), and only 4 had a score below 21.

The time of protective ileostomy (divided into up to 6 weeks, between 6 and 12 weeks, and over 12 weeks) was statistically correlated with the appearance of LARS, with the LARS rate of occurrence being higher in patients with a longer ileostomy time (*p*-value 0.0081). Indication for the closure of the ileostomy was related to the normal colonoscopic appearance of the anastomosis at 6 weeks in patients without indication for adjuvant chemotherapy. The closure of the ileostomy performed after oncological treatment was completed in patients with advanced disease (higher pTMN), and closure after a normal colonoscopic examination was performed in patients with anastomotic complications.

The correlation coefficient between initial LARS and LARS at 6 months was calculated; we obtained a r-value equal to 0.8 (very high positive). This finding may be explained by the fact that cases with high values of initial LARS (over 30) also have increased values 6 months after the diagnosis, while cases with low values of initial LARS also have a low LARS at 6 months. We also calculated the Pearson correlation coefficient between the LARS score at 6 months and 2 years, with a value of 0.5 (good), which can be explained by the fact that 23 of 36 patients with high initial LARS scores showed minor LARS at 6 months, but only 3 patients had LARS symptoms at 2 years. Of the 61 patients diagnosed at discharge, 36 had major LARS (subgroup 1), and 25 had minor LARS (subgroup 2). In subgroup 1 (major LARS), the 6-month assessment showed that 14 patients out of 36 switched to minor LARS (score between 21–30), 14 patients had a score lower than 20 (non-LARS), and 8 patients remained in the major LARS category. At 2 years, four patients regressed to minor LARS, and four patients remained within major LARS classification, but seven out of eight patients with major LARS at 6 months requested terminal colostomy. In subgroup 2, at 6 months, 23 patients had a score of non-LARS (below 21), although complete remission of symptoms (score 0) was recorded only in 2 patients; at 2 years, a significant improvement was noted: Only three patients still had LARS-specific symptoms but with a score less than 10.

## 4. Discussion

LARS is a common complication of patients with rectal cancer and anterior low rectal resection. With the development of devices for mechanical suturing and trans-anal descent techniques, the margin of rectal resection compared to anal verge has decreased (up to 2 cm of OAE), and consequently, the incidence of permanent colostomy after rectal amputations has likewise decreased. Despite this, a group of patients will still develop symptoms that significantly influence their quality of life (later classified as LARS) [1,2,3]. In 2012, a score that noted symptoms after rectal resection whose value fits the pathology of one of the two subclasses of LARS was validated in Denmark [10]; a similar score is used today in Romania [11]. When LARS symptoms occur, a therapeutic plan developed immediately postoperatively by surgeons, oncologists, and physiotherapists can sometimes alleviate the symptoms over time. Identification of early symptoms, monitoring, follow-up, and rapid establishment of a treatment algorithm leads to satisfactory symptom control, thus increasing the probability of improvement in severe impairment of quality of life (QOL) [1].

The proportion of LARS in the present sample was similar to that in other studies [2,4,12]; in our study, 58% had a score greater than 21, 19.5% had a score of 0, and 22 patients (21.5%) had symptoms specific to LARS but achieved a score less than 20. The first bowel movements after surgery oriented the surgeon towards a possible LARS diagnosis; the use of a diaper (Pampers) at the periodic visit was suggestive of LARS even before filling out a questionnaire, so the need for a diaper (Pampers) was highlighted in all patients with LARS and only in 22 patients classified as non-LARS.

Patients who develop at least one of the functional disorders summarized under the term low anterior resection syndrome (LARS) may have single or multiple risk factors. In our study, diabetes mellitus (type 1 and 2) and obesity (GMI greater than 30 kg/m^2^) were statistically positively correlated with a high LARS score, and this finding was similar to that of other published studies [1,12]. The mechanism for the association between diabetes and LARS has been described [1], but the contribution of obesity to an increased rate of LARS is not fully elucidated, although the risk of perioperative complications (including anastomotic) is higher in obese patients. In our study, LARS was more frequent in males in comparison with women (*p*-value 0.03); the literature shows conflicting data [13,14,15,16].

LARS frequency was correlated to the type of resection (imposed by tumor location), with higher rates in patients with low resection and total mesorectal excision in comparison with other interventions (*p*-value 0.00001), although it also occurs in medium rectal resections; the data were similar to that of other studies [1,2,13,14,15,16]. Tumors located in the upper rectum can benefit from a partial excision of the mesorectum (limited at 4 cm below the tumor), while in cancers of the middle and lower rectum, rectal resection and total excision of the mesorectum are indicated [17]. Anterior rectal resection with total excision of the mesorectum will cause a loss of reservoir function (the rectal ampulla’s large storage capacity is replaced by a colon segment with a smaller diameter) and a markedly disturbed outlet (the longitudinal fibers are arranged in the form of a layer) in contrast to the banded arrangement of the colon, leading to a significant impairment of rectal compliance [18], with an increase in false sensations of defecation. This was demonstrated by Karanijia and colleagues, who showed a worsening of rectal ampulla function proportional to a reduced distance of the anastomosis anal margin [1,18] contributing to the loss of reservoir function, squeezing, and evacuation. This view explains the summaries of our study and the finding that a remaining lower rectum greater than 4 cm in length was associated with significantly better functional outcomes (recto anal inhibition reflex, rectal capacity, etc.) compared to patients who had less than 4 cm of remaining rectum [19,20]. Although the literature data are clear, our study and other reports have shown that a small proportion of patients with partial excision of the mesorectum associated with a previous rectal resection may develop LARS. One possible explanation is that in cancers of the upper rectum, ligation of the pedicle of the inferior mesentery is performed at the origin, thus involving denervation of the parasympathetic and sympathetic nervous system, which influence the activity of the colon by increasing the negative effects in regards to rectal discharge [1,21,22]. At the same time, the introduction of the concept of “rectosigmoid brake”, which plays an important role in sigmoid motility by delaying rectal filling, is another explanation for the occurrence of LARS in these patients [1,23,24].

Neoadjuvant radiotherapy may be a risk factor for LARS [1]; however, in our study, this factor did not significantly influence the occurrence of LARS, especially in patients with locoregionally advanced tumors who regressed significantly after major LARS oncological treatment. The mechanisms involved in the appearance of post-radiotherapy LARS are complex and are attributed to inflammation of the pelvis, rectal wall, and mesorectum [25] or to secondary neuropathy [26], especially in patients with radiotherapy performed for tumor downstaging.

The type of colorectal anastomosis (mechanical or manual) performed by the surgeon is influenced by the distance from the anal verge, with mechanical anastomosis being easier to perform technically compared to manual anastomosis after total excision of the mesorectum. The addition of other factors, such as obesity, intraoperative bleeding, or the occurrence of other incidents that increase the operative time, can influence the surgeon in the choice of mechanical anastomosis. In our study, there was a statistically significant correlation between the type of anastomosis and the emergence of LARS (*p*-value 0.007), but these data are influenced by the type of ultra-low resection and the need to use staplers to restore digestive transit (performing a manual anastomosis is difficult or impossible to perform from a technical point of view). On the other hand, the appearance of an anastomotic complication of the stenosis or disjunction clinically expressed by pelvic abscess or fistula) significantly increased (*p*-value 0.04) the risk of developing major LARS and terminal colostomy [27]. Of the 102 patients included in the study, 14 patients (13.7%) had complications during the evolution, and because of major LARS, colostomy was performed in 8 cases (7.8%).

The indication for temporary protective ileostomy depends on the distance between the anastomosis and the anal verge and also on the experience of the surgical team, which has a role in the prophylaxis of infectious complications secondary to a disjunction of the anastomosis [28]. The time of ileostomy depends on the occurrence of postoperative anastomotic complications, the need for adjuvant chemotherapy required by the final staging of the disease (pTNM), and the availability of the surgeon and the patient, although a consensus does not exist [29]. As in another report [30], in our study, the increase in the duration of ileostomy for different reasons was statistically significantly correlated with the occurrence of LARS (*p*-value 0.0081), while an ileostomy over 12 weeks was correlated with a higher LARS score. The lack of pelvic floor muscle usage for a prolonged time [31] and changes in the colonic environment by modifying the composition of the microbiota [32] could explain the positive correlation between the duration of an ileostomy and the appearance of LARS. Although there are studies on the establishment of physical therapy in patients with anterior lower rectum resection [33,34], the indication for physical therapy was also established in the patients included in our study, but the follow-up for that purpose was difficult to perform because of early quitting, especially in case of elderly patients.

The reassessment of patients with LARS syndrome at 6 months and 2 years showed a decrease in LARS score in many cases, although some patients can have a similar score even at 2 or 4 years [35]. Patients with a high score at discharge have an average lower score at 6 months and 2 years, although they remain in the LARS category. Patients with minor LARS at discharge presented a greater decrease in the score at 6 months and 2 years, with some entering a non-LARS category. Similar to another study [36], an improvement in symptoms at 2 years was generally noted; a multimodal approach may be the best option for management for patients with LARS. Diet, medications, training for the pelvic floor muscle (PFMT), and rectal irrigation can be added as secondary treatments [37]. The classification of major LARS at 6 months was associated with a significant rate of definitive colostomy (87.5%).

The therapeutic management of patients with LARS should address all potential contributing factors that may converge in the etiology of this syndrome. First-line interventions include dietary modifications, pelvic floor muscle training, and biofeedback [2,38]. If symptoms persist beyond one year, rectal irrigation and sacral nerve stimulation may be considered as adjuvant therapies; colostomy may be indicated in cases of multimodal treatment failure or upon patient request [37].

The limitations of this study arise from its retrospective nature, as it was conducted in a single department, with limited follow-up time (2 years), and with a relatively small number of patients with LARS syndrome included. The small sample size did not adequately permit an in-depth analysis of risk factors for LARS symptoms development and also for every symptom included in the definition of LARS. Additionally, the inclusion of only patients undergoing open surgery, without considering laparoscopic or robotic procedures, represents a limitation of this study. However, it opens another chapter in understanding the development of this syndrome in patients undergoing minimally invasive surgeries. By incorporating data from multiple centers, it may be possible to analyze a larger number of cases, thereby increasing the ability to draw statistically significant conclusions. Furthermore, a longer follow-up period or a prospective study of patients with LARS symptoms can be more accurate in predicting the role of risk factors for LARS appearance and also in evaluating the importance of classical and alternative treatments for LARS.

The perspective of patients on the development of LARS is a very important factor because LARS symptoms have a major impact on the quality of life. From that point of view, a future study that incorporates the impact of LARS therapy on the quality of life may be useful for refining the therapeutic algorithm.

## 5. Conclusions

LARS syndrome is a progressive complication that appears after multimodal cancer therapy for rectal cancer and that affects the patient’s life outside the neoplasia itself. The patient’s need for diapers in the postoperative period in the hospital is the first sign of possible LARS syndrome. Obesity, size of the remaining rectum, total excision of the mesorectum, anastomotic complications, and prolonged ileostomy time are cooperating factors in the etiology of LARS. Our study showed a trend toward improving the LARS category from major to minor LARS, although, in a small number of patients, the improvement was insignificant, and these patients finally ended up needing permanent colostomy. The persistence of major LARS at 6 months after surgery may predict in a higher proportion the need for a definitive colostomy.

## Figures and Tables

**Figure 1 cancers-16-04175-f001:**
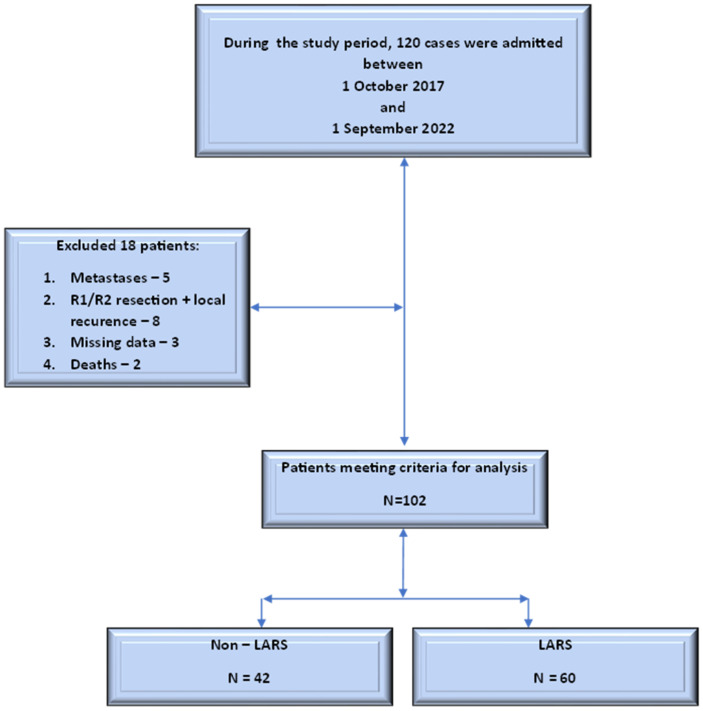
Flow diagram of the patient-selection process.

**Figure 2 cancers-16-04175-f002:**
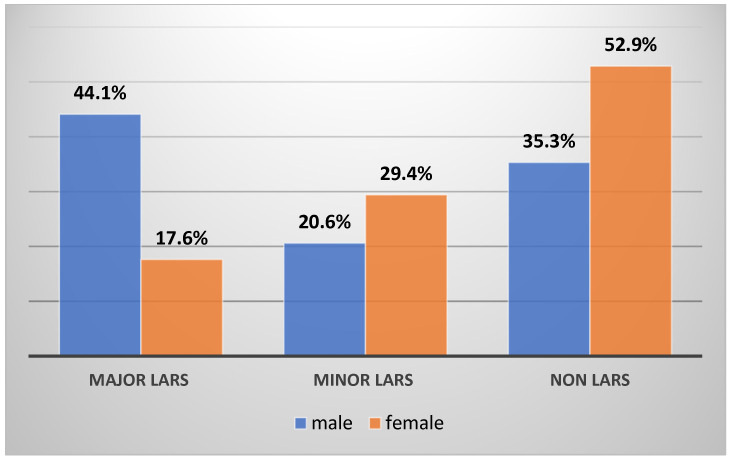
Gender distribution of the patients.

**Figure 3 cancers-16-04175-f003:**
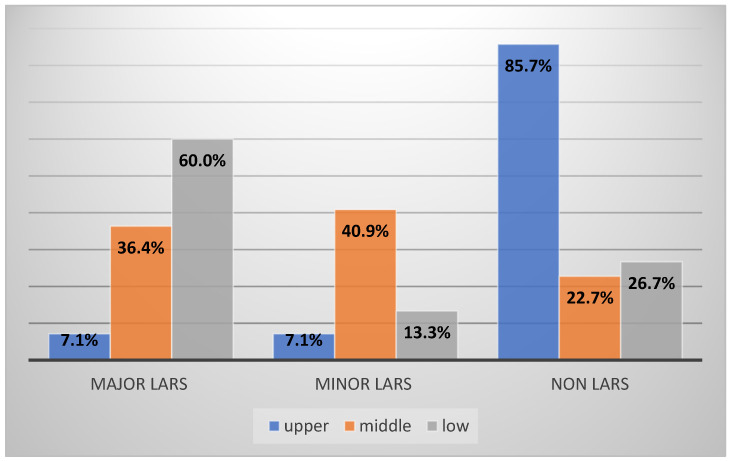
LARS syndrome distribution according to tumor topography.

**Figure 4 cancers-16-04175-f004:**
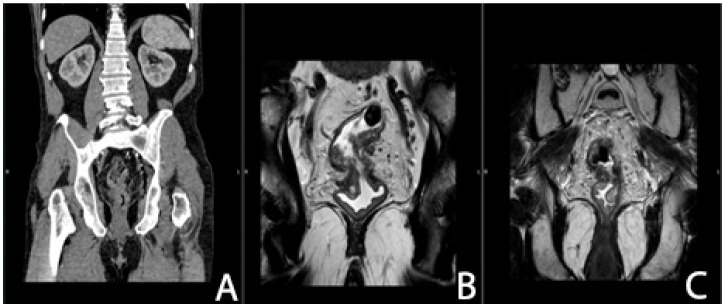
(**A**–**C**) CT examination (**A**) and MRI (**B**,**C**) in a patient with rectal cancer before radiotherapy: tumor of the upper-middle rectum with extension to the mesorectum and pelvic peritoneum cT4aN2M0 (frontal plane). The images are used with the permission of and were provided courtesy of C.V.O. and D.N.F., respectively.

**Figure 5 cancers-16-04175-f005:**
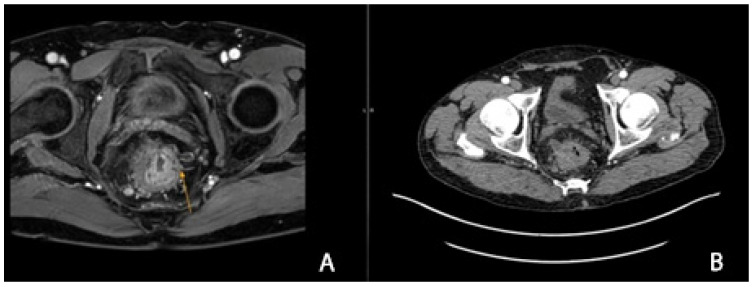
(**A**,**B**) MRI examination: rectal cancer before radiotherapy: upper-middle rectum with extension to the mesorectum and pelvic peritoneum cT4aN2M0 (yellow arrow). The images are used with the permission of and were provided courtesy of C.V.O. and D.N.F., respectively.

**Figure 6 cancers-16-04175-f006:**
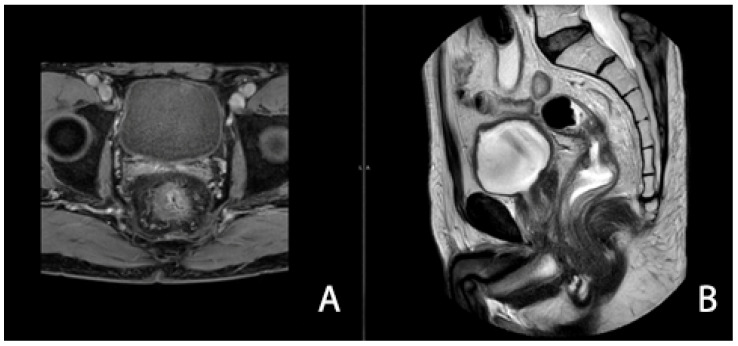
(**A**,**B**) MRI examination after radiotherapy reveals the regressive appearance of the tumor and peritumoral lymphadenopathy. The images are used with the permission of and were provided courtesy of C.V.O. and D.N.F., respectively.

**Table 1 cancers-16-04175-t001:** The LARS score, modified after [7].

LARS—Score—Scoring instructions Add the Scores from Each 5 Answers to One Final Score	
**Do you ever have ocasions when you cannot control your flatus (wind)?** ○ No, Never ○ Yes, less than once per week ○ Yes, at least once per week	**0** **4** **7**
**Do you ever have any accidental leakage of liquid stool?** ○ No, Never ○ Yes, less than once per week ○ Yes, at least once per week	**0** **3** **3**
**How often do you open your bowels?** ○ More often than 7 times per day (24 h) ○ 4–7 times per day (24 h) ○ 1–3 times per day (24 h) ○ Less than once per day (24 h)	**4** **2** **0** **5**
**Do you ever have to open your bowels again within one hours of the last bowel opening?** ○ No, Never ○ Yes, less than once per week ○ Yes, at least once per week	**0** **9** **11**
**Do you ever have such a strong urge to open your bowels that you have to rush to the toilet?** ○ No, Never ○ Yes, less than once per week ○ Yes, at least once per week	**0** **11** **16**
**Total score:**	
**Interpretation** **0–20 No LARS** **21–29 Minor LARS** **30–42 Major LARS**	

**Table 2 cancers-16-04175-t002:** Clinicopathological characteristics of patients with LARS syndrome and non-LARS.

Parameter	Total	Major LARS	Minor LARS	Non-LARS	*p*-Value
**No. of cases**	102	36 (35.3%)	24 (23.5%)	42 (41.2%)	
**Gender %**					0.030 ^(a)^*
Female	34 (33.3)	6 (17.6)	10 (29.4)	18 (52.9)	
Male	68 (66.7)	30 (44.1)	14 (20.6)	24 (35.3)	
**Age**		58 + 13.5	58.7 + 12	63.9 + 12.9	0.110 ^(b)^
**Tumor location (%)**					
Upper rectal	28 (27.4)	2 (7.1)	2 (7.1)	24 (85.7)	<0.0001 ^(a)^*
Middle rectal	44 (43.2)	16 (36.4)	18 (40.9)	10 (22.7)	
Low rectal	30 (29.4)	18 (60)	4 (13.3)	8 (26.7)	
**Radiotherapy (%)**					
Yes	83 (81.4)	33 (91.6)	19 (82.1)	29 (75)	0.086 ^(a)^
No	19 (18.6)	3 (8.4)	5 (17.9)	11 (25)	
**Chemotherapy (%)**					
Yes	88 (86.3)	32 (89.9)	20 (79.2)	36 (85.7)	0.821 ^(a)^
No	14 (13.7)	4 (11.1)	4 (20.8)	6 (14.3)	
**Type of surgery (%)**					
LAR + PME	22 (21.6)	1 (2.7)	1 (4.2)	20 (47.6)	<0.001 ^(a)^*
LAR + TME	12 (11.8)	4 (11.1)	4 (16.7)	4 (9.5)	
ULAR + TME	68 (66.6)	31 (86.1)	19 (79.1)	18 (42.9)	
**Ileostomy (%)**					
Yes	68 (66.6)	32 (88.9)	18 (75)	20 (47.6)	<0.001 ^(a)^*
No	34 (33.3)	4 (11.1)	6 (25)	22 (52.4)	
**Complication (%)**					
Stenosis	4 (3.9)	4 (11.1)	-	-	
Fistula	10 (9.8)	6 (11.1)	2 (8.3)	2 (4.8)	
None	88 (86.3)	26 (72.2)	22 (91.7)	40 (95.2)	
**Duration of ileostomy (%)**					
Without	26 (25.5)	-	6 (25)	20 (47.6)	-
6 weeks	26 (25.5)	10 (27.8)	4 (16.7)	12 (28.6)	0.092 ^(a)^
7–12 weeks	44 (43.1)	20 (55.6)	14 (58.3)	10 (23.8)	
12+ weeks	6 (5.9)	6 (16.7)	-	-	-
**LARS at 6 months**	60	6	14	40	
**LARS at 2 years**	35	7	14	14	
**Colostomy**	7	4	3		
**Type of anastomosis (%)**					
Manual	20 (19.6)	2 (5.5)	4 (16.7)	14 (33.3)	0.007 ^(a)^*
Mechanical	82 (80.4)	34 (94.5)	20 (83.3)	28 (66.7)	
**Comorbidities**					
Obesity	54 (58.7)	28 (56)	14 (58.3)	12 (66.7)	<0.001 ^(a)^*
Diabetes mellitus	38 (41.3)	22 (44)	10 (41.7)	6 (33.3)	<0.001 ^(a)^*
**pT (%)**					
**yT_0_**	10 (9.8)	2 (5.5)	2 (8.3)	6 (14.3)	0.112 ^(a)^
**T_1_**	8 (7.8)	2 (5.5)	4 (16.6)	2 (4.7)	
**T_2_**	46 (45.1)	14 (39)	14 (58.4)	18 (43)	
**T_3_**	38 (37.3)	18 (50)	4 (16.7)	16 (38)	
**pN (%)**					
**N_0_**	36 (35.3)	18 (50)	6 (25)	12 (37.5)	
**N_1_**	42 (41.2)	8 (22.2)	10 (41.7)	14 (43.7)	
**N_2_**	22 (21.6)	8 (22.2)	8 (33.3)	6 (18.8)	
**N_3_**	2 (1.9)	2 (5.6)	-	-	

* statistically significant (*p* < 0.05); ^(a)^ chi-square test; ^(b)^ ANOVA test; LAR: low anterior resection; U-LAR: ultra-low anterior resection: PME: partial excision of the mesorectum; TME: total mesorectal excision.

## Data Availability

The data presented in this study are available on request from the corresponding author.

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
