# Peer review of "The Diagnosis and Evolution of Patients with LARS Syndrome: A Five-Year Retrospective Study from a Single Surgery Unit"

_cancers, 2024, doi:10.3390/cancers16244175_

Round 1
Reviewer 1 Report
Comments and Suggestions for Authors
Attached you will find an extensive revision of the paper. Below are general comments.
Introduction
- LARS is mentioned, but the definition of LARS is not explained, nor is there a reference provided for it.
- It is stated that a significant percentage of patients develop LARS syndrome, but no numerical data or articles from the literature are used to support this.
- The introduction does not mention which factors found in the literature contribute to the development of LARS.
Results
Figure 1
- One starts with 120 cases admitted between [dates]. But how many patients were originally eligible for this study? How many patients met the inclusion and exclusion criteria?
Table 1, Figure 2, 3, 7
- You make incorrect interpretations by comparing absolute numbers of groups with unequal numbers of individuals.
Discussion
- Strengths, limitations, and future research are not included.
General
- Refer in text to figures.
- Ensure there are sufficient references in the text.
- Avoid vague language, e.g., "some of the patients.

I have made comments regarding the English in the attachment.
Author Response
Thank you for your observations. Please see the attachment!

Reviewer 2 Report
Comments and Suggestions for Authors
Positive Aspects
- Relevant Clinical Focus:
- The study addresses Low Anterior Resection Syndrome (LARS), a condition that significantly impacts quality of life for rectal cancer patients. Given the increasing prevalence of LARS post-surgery, the study’s clinical focus on identifying risk factors and symptom progression is valuable and relevant for healthcare providers.
- Comprehensive Data Collection:
- The manuscript covers a wide range of variables, including demographic data, tumor characteristics, comorbidities, and postoperative complications. This broad data collection provides a holistic view of factors that may influence the development of LARS and helps paint a clearer picture of its etiology.
- Detailed Methodology:
- The methodology section is well-documented, outlining inclusion and exclusion criteria, data collection methods, and statistical analyses. This clarity allows for reproducibility and makes it easier for other researchers to follow the study’s process, which adds credibility.
- Statistically Significant Findings:
- The study identifies statistically significant correlations, such as the association between longer ileostomy duration and higher LARS rates. These findings could be useful in guiding future research or informing preoperative counseling for patients about the risk factors for LARS.
- Useful Insights into Patient Demographics:
- By analyzing factors like obesity, diabetes, gender, and tumor location, the study sheds light on specific patient subgroups that may be at higher risk of developing LARS. These insights can assist in personalized patient management and targeted interventions.
- Application of Diagnostic Scoring:
- The study uses the LARS score to classify patients' symptom severity. This scoring system provides a standardized method of assessing symptom burden, which enhances the study’s rigor and enables comparisons with other research.
- Illustrative Imaging:
- The inclusion of CT and MRI images adds a visual component, helping readers better understand the anatomical and pathological changes involved. This also highlights the diagnostic process and the use of imaging as part of the patient assessment.
- Longitudinal Assessment of Symptoms:
- The study’s retrospective design, spanning five years, allows for an analysis of symptom evolution over time. Tracking LARS symptoms at 6 months and 2 years provides useful insights into the progression and potential improvement of symptoms, adding depth to the findings.
- Ethical Considerations:
- The manuscript clearly states that ethical approval was obtained and describes steps taken to protect patient confidentiality, which strengthens the study’s credibility and adherence to research standards.
- Foundation for Future Research:
- By exploring multiple risk factors and outcomes related to LARS, the study lays a foundation for future research on LARS management and prevention. It also highlights the importance of early symptom identification and the need for further investigation into long-term treatments.
Negative Aspects
- Lack of Cohesive Structure and Clarity in Results:
- The manuscript’s structure feels somewhat disjointed, particularly within the results section. Findings are presented without clear transitions or summaries, which makes it difficult for readers to see overarching trends or significant patterns across cases. Presenting key takeaways upfront or summarizing each subsection could help in synthesizing findings.
- Limited Statistical Depth:
- Although the study mentions various statistical correlations, it lacks depth in terms of interpretation. The analysis seems surface-level, primarily limited to p-values without further contextual insights. There’s little discussion on the clinical relevance or practical implications of these findings. Enhancing the statistical analysis by including effect sizes or discussing the limitations of each correlation would add rigor.
- Redundancy in Reporting:
- There is significant repetition in discussing factors associated with LARS, such as obesity, rectum size, and the duration of ileostomy. These factors are refined across sections without adding new insights, which dilutes the impact of the findings and leads to redundancy. Consolidating this information to highlight each factor's unique contribution would help avoid repetitive content.
- Insufficient Focus on Long-term Outcomes:
- While the study mentions that patients were monitored for two years, it provides limited insight into the trajectory of symptoms beyond this timeframe. Given the chronic nature of LARS, discussing longer-term outcomes or referencing literature on outcomes beyond two years would make the findings more comprehensive and relevant.
- Inadequate Consideration of Confounding Variables:
- The manuscript briefly mentions demographic data and comorbidities, but there’s no clear attempt to control or account for potential confounding variables, such as age, underlying health conditions, or lifestyle factors. These variables could influence both the occurrence and severity of LARS symptoms. A more thorough analysis that adjusts for these confounders would strengthen the study’s validity.
- Weak Discussion of Alternative Treatments:
- While the study describes dietary and physical therapy interventions, there is minimal discussion on other treatment modalities for LARS. Addressing more comprehensive treatment options, such as pharmacological interventions or surgical alternatives, would improve the practical applicability of the research findings for clinicians.
- Limited Generalizability Due to Sample Size and Single-center Design:
- The study’s single-center, retrospective design and relatively small sample size limit the generalizability of its findings. There is no discussion on how the local population's characteristics or surgical practices may differ from other settings. Acknowledging these limitations explicitly and recommending multi-center studies would make the manuscript more scientifically robust.
- Over-reliance on Retrospective Data:
- The study relies solely on retrospective data, which introduces recall bias and limits the control over data quality. There’s no attempt to mitigate these biases, such as by cross-validating findings with prospective data where possible. Incorporating a prospective element or recommending one for future research would enhance the study’s reliability.
- Ambiguity in Data Presentation:
- The tables and figures lack clarity, with some parameters grouped in ways that make interpretation challenging. For instance, combining minor and major LARS scores without clear explanations on specific symptoms can obscure the variability in symptom presentation. Revising data presentations to show clearer breakdowns and adding labels for easy comparison would improve readability.
- Missed Opportunity for a Patient-centered Perspective:
- Although the study examines factors contributing to LARS, there is limited discussion of patient-reported outcomes (PROs) or quality-of-life measures, which are essential for understanding the impact of LARS on patients. Incorporating PROs or emphasizing the need for such data in future studies would underscore the study’s relevance to patient-centered care.
Suggestions for Improvement
- Enhance Cohesion: Consolidate repetitive information and introduce clear transitions to enhance narrative flow and readability.
- Strengthen Statistical Analysis: Expand statistical methods beyond p-values and interpret the clinical relevance of findings.
- Highlight Practical Implications: Add more detail on alternative interventions for LARS and the implications of each treatment pathway.
- Control for Confounding Variables: Re-analyze the data to account for confounders like age and comorbidities or discuss these limitations.
- Expand Discussion on Generalizability: Address limitations from single-center data and propose multi-center studies for broader insights.
Author Response
Thank you for your extensive and precise observations. For the response, please see the attachment!

Reviewer 3 Report
Comments and Suggestions for Authors
I have no additional questions for the authors.
Author Response
Thank you for your observations!
Round 2
Reviewer 1 Report
Comments and Suggestions for Authors
The percentages in Figures 2 and 7 are still incorrect, which will lead the reader to misinterpret gender distribution and anastomotic complications.
Figure 8: You make incorrect interpretations by comparing absolute numbers of groups with unequal numbers of individuals.

Author Response
Page 2 Introduction
- Thank you for your observations. We tried to make the introduction more cohesive, but we wanted to preserve the quality of the information offered in the Introduction section.
Page 5. Table 2
- We modified the data.
Page 7.
- We modified the figure according to the data from the table.
Page 9
We removed figures 7 and 8
Page 12.
- We added the references.

Reviewer 3 Report
Comments and Suggestions for Authors
I have no additional questions for the authors
Author Response
Thank you for your comments!
Round 3
Reviewer 1 Report
Comments and Suggestions for Authors
If you want to compare tumor height with the percentage of patients having major, minor and no LARS, you need to ensure that the total for major LARS is 100%, minor LARS is 100% and no LARS is 100%. For Figure 2, this has already been corrected. In Table 2, you also need to apply this for all your parameters, not just for gender.
Author Response
Comment: If you want to compare tumor height with the percentage of patients having major, minor and no LARS, you need to ensure that the total for major LARS is 100%, minor LARS is 100% and no LARS is 100%. For Figure 2, this has already been corrected. In Table 2, you also need to apply this for all your parameters, not just for gender.
Response:
We modified the percentages for the tumor location line in table 2 according to your suggestion of and also we have replaced figure 3 according to the new data. The percentages were calculated in the same manner as for gender (and figure 2).
Best Regards!